# Bridging the Scientific Gaps to Identify Effective Treatments in Adrenocortical Cancer

**DOI:** 10.3390/cancers14215245

**Published:** 2022-10-26

**Authors:** Nicholas Michael, Naris Nilubol

**Affiliations:** Surgical Oncology Program, Center for Cancer Research, National Cancer Institute, National Institutes of Health, Bethesda, MD 20892, USA

**Keywords:** adrenocortical carcinoma, genomics, transcriptomics, adrenalectomy, immunotherapy, quantitative high throughput drug screening

## Abstract

**Simple Summary:**

Adrenocortical cancer (ACC) is a rare malignancy with a poor prognosis and few treatment options. Surgery is the only curative option, but many patients present after the disease has become metastatic or recurs after surgery. There have been few advances in treatment since mitotane’s discovery in the 1960s, and it remains the only FDA-approved ACC treatment. This paper reviews the current treatment approaches, what we have recently learned about the disease, and the next steps needed to advance research and improve therapeutic options for this disease.

**Abstract:**

Adrenocortical cancer (ACC) typically presents in advanced stages of disease and has a dismal prognosis. One of the foremost reasons for this is the lack of available systemic therapies, with mitotane remaining the backbone of treatment since its discovery in the 1960s, despite underwhelming efficacy. Surgery remains the only potentially curative option, but about half of patients will recur post-operatively, often with metastatic disease. Other local treatment options have been attempted but are only used practically on a case-by-case basis. Over the past few decades there have been significant advances in understanding the molecular background of ACC, but this has not yet translated to better treatment options. Attempts at novel treatment strategies have not provided significant clinical benefit. This paper reviews our current treatment options and molecular understanding of ACC and the reasons why a successful treatment has remained elusive. Additionally, we discuss the knowledge gaps that need to be overcome to bring us closer to successful treatment and ways to bridge them.

## 1. Introduction

The first descriptions of adrenocortical cancer (ACC) come from autopsy studies in the 1800s. At the time, the only method for diagnosing an adrenal mass in a living patient was operative exploration, and this was only performed when the mass had become large enough that it was palpable on physical exam. Physicians recognized early on that these patients had an abysmal prognosis and that surgery was the only potentially curative treatment [1]. At the time, adrenalectomy was a morbid procedure, and it did not offer a benefit to these patients who were presenting with extensive tumors and advanced disease. Adding to the morbidity of resection in the early 20th century was the inability to replace steroids in those who developed postoperative adrenal insufficiency following adrenalectomy in cortisol-producing ACC. The first breakthrough in the treatment of ACC came in the 1940s when steroid replacement became available. Prior to this, 80% of patients with ACC and hypervirulization who underwent adrenalectomy quickly passed away in the postoperative period, some within hours, due to adrenal insufficiency [1,2].

The next step forward in ACC treatment came with the discovery of mitotane. As a derivative of the insecticide dichlorodiphenyltrichloroethane (DDT), mitotane was found to cause adrenal atrophy in animal studies in the 1950s leading to the first clinical report of its use in 1960. This report showed evidence of tumor regression in 7 out of 18 patients with metastatic ACC treated with mitotane, leading the National Cancer Institute to further investigate its use [3]. Mitotane subsequently became the only FDA-approved systemic treatment for ACC and remains so today. Mitotane has a narrow therapeutic window and requires frequent drug-level monitoring, and it is associated with frequent treatment-related toxicities [3,4]. The anti-tumor efficacy of mitotane is low as the response rate is approximately 20–30% [5]. Nevertheless, mitotane is commonly used to control Cushing’s syndrome in patients with cortisol-producing tumors and for patients with high-risk, persistent, or recurrent ACC. A better alternative has not been found in the 60 years since mitotane’s discovery, and it is still a key part of the treatment algorithm for advanced ACC [1,2,6].

The lack of new and more effective treatments is partly due to the difficulties of studying such a rare disease. The few epidemiological studies on ACC estimate the incidence in adults at one case/million/year [4,7,8]. In children it is even more rare (except in Southern Brazil, as discussed below), with an incidence of 0.3 cases/million/year in the United States [9]. Despite the increasing frequency of abdominal imaging and incidentally found tumors, about half of patients still present with stage III or IV disease [4]. A total of 60% of adult patients present with hypercortisolemia, which is an independent predictor of poor prognosis [10]. For those patients who undergo successful initial surgical treatment, the recurrence rate is at least 50% despite an R0 resection. The prognosis of patients with stage IV ACC remains poor as five-year overall survival (OS) is approximately 15% [10,11]. Pediatric patients typically have better outcomes in early-stage disease, but once metastatic, five-year OS is similarly around 15% [9,12,13,14]. The poor prognosis of advanced ACC has not changed in decades due to the lack of effective systemic treatments, and meaningful clinical benefits have not been shown in any of the clinical trials using novel therapeutics in advanced ACC [15,16,17]. In this review, we discuss the current state of research and treatment for advanced ACC, the knowledge gaps that need to be overcome, and how they can be addressed.

## 2. Current Understanding of ACC

Because insight into the molecular pathology in ACC is crucial to discovering novel and effective therapeutics, several landmark studies discussed below have greatly improved our understanding of the genetic and molecular alterations leading to tumor initiation and progression. The profiling of the genomic and molecular landscape, including the analysis of mutations, chromosomal and copy number abnormalities, methylation profiles, mRNA, and micro-RNA (miRNA) dysregulation, has become possible over the past few decades and is being applied widely in oncology research. ACC is no exception. The Cancer Genome Atlas (TCGA) project and the European Network for the Study of Adrenal Tumors (ENS@T) have conducted studies with relatively large cohorts on all stages of the disease in adults [18,19]. More recently, whole genome analysis has also been performed on smaller cohorts of patients with metastatic disease [20,21].

Mutation analysis in both the TCGA and ENS@T studies confirmed that the mutational burden for ACC is low. In the ENS@T cohort, there were 0.6 mutations per megabase (Mb) and 0.9 mutations/Mb in the TCGA cohort, compared to an average of 3.6 mutations/Mb across all adult malignancies [18,19,22]. The Wnt-signaling (*CTNNB1*, *ZNRF3*) and p53-Rb (*TP53*, *Rb*, *CDKN2A*, *RPL22)* pathways were already known to play a role in ACC development, with the activation of the Wnt-signaling pathway thought to be one of the initial events in tumor development [23]. This is not the case in pediatric ACC. In a study of the genomics of pediatric ACC by Pinto et al., alterations in Wnt-signaling were only found in 8% of cases [24]. Cell cycle dysregulation through the p53-Rb pathway is widespread across cancers, and it is not surprising that it is also a culprit in ACC. The ENS@T and TCGA studies confirmed that the Wnt-signaling and the p53-Rb pathways were the most frequently altered by mutations, copy number variants, or epigenetic changes, with 41% and 45% of tumors affected, respectively, in the TCGA study [18,19,20,21]. More recently, a study using next-generation sequencing (NGS) of over 100 ACC samples found CDK4 to be the most frequent mutation, appearing in 43% of cases [25].

*TP53* mutations are particularly important in pediatric ACC. Germline mutations in *TP53* cause Li–Fraumeni syndrome, a rare autosomal dominant syndrome that leads to the development of multiple cancer types, including ACC [26]. There are numerous *TP53* mutations that can lead to cancer development which have variable penetrance. A notable example is the R337H mutation, which is common in southern Brazil with a prevalence of 0.3%. In this small region of the world, the incidence of pediatric ACC is about 15-fold higher than the rest of the world at about four cases/million/year [27]. Pinto et al. found that *TP53* mutations were identified in 76% of pediatric cases overall, and even when the R337H mutation from southern Brazil were excluded, other *TP53* mutations were found in 64% of cases [24].

Genes related to the development of other hereditary syndromes, including *IGF2*, *MEN1*, *NF1*, and *PRKAR1A*, were also frequently altered [18,19,20,21]. *IGF2*, one of the genes associated with Beckwith–Wiedemann Syndrome (BWS) and known to be over-expressed frequently in both pediatric and adult ACC, was over-expressed in 86% of tumors in the TCGA study [19]. In the pediatric study by Pinto et al., a loss of heterozygosity at 11p15, the region where *IGF2* is located, was identified in 91% of cases. This number excluded two patients with BWS, who have 11p germline homozygosity [24]. The IGF pathway was a promising treatment target as it is a common alteration in ACC, but, as discussed below, this strategy was not effective [28].

Mutations in *MEN1* cause the Multiple Endocrine Neoplasia type 1 (MEN1) syndrome. Adrenal lesions frequently occur in MEN1, but the rate of ACC in these adrenal lesions is about 5%. Menin’s role in ACC development has yet to be clarified [29]. *PRKAR1A* encodes a protein kinase regulator and is the gene responsible for the development of the Carney complex, another rare endocrine neoplasia syndrome associated with ACC development [30]. *NF1* is a tumor suppressor gene through the down-regulation of the RAS pathway, and mutations lead to neurofibromatosis type 1 (NF1), characterized by the development of multiple benign and malignant tumors. However, no definite association between ACC and NF1 is known, although multiple case reports of NF1 patients that developed ACC have been published [31]. The relationship between these genes and ACC tumorigenesis is unclear; they may point to the involvement of critical signaling pathways for subsets of patients, but there are no clearly defined roles at this point.

Lynch Syndrome is another hereditary cancer syndrome caused by defective DNA mismatch repair and is present in about 3% of patients with ACC [32]. In the TCGA and ENS@T study cohorts, no single gene involved in mismatch repair, or DNA damage repair in general, was found with significant frequency [18,19]. In the studies from Fojo et al. and Lavoie et al., however, genes involved in DNA repair, including mismatch repair genes (*MLH1*, *MSH6*) as well as *BRCA1*, *BRCA2*, and others, were altered in around half of the tumors [20,21].

Other genes harboring recurrent and pathogenic somatic mutations, deletions, or amplifications in ACC included *DAXX*, *ATRX*, *MLL4*, *TERT*, and *TERF2. TERT* is an essential component of the telomerase complex that is over-activated in many cancer types as the primary mechanism for maintaining telomere length as these cells replicate. *TERF2* is part of the Shelterin complex that helps maintain telomere length through interactions with telomerase and was amplified in 7% of tumors in the TCGA study [19,33]. In all of the studies mentioned above, *TERT* had altered expression in a significant number of tumors analyzed, including amplification in over 70% of patients in the cohort from Fojo et al. [20]. In the TCGA analysis, *TERT* over-expression occurred frequently with whole-genome doubling (WGD) events. WGD is common across cancer types associated with genomic instability, increased chromosomal aberrations, and more frequent mutations [19,34]. It is thought that increased *TERT* expression is required to maintain adequate telomere length for cell survival in the presence of the instability caused by WGD [19].

*DAXX* and *ATRX* form a well-known complex responsible for histone deposition at telomeres and pericentric heterochromatin. [35]. When this complex is dysfunctional, it can lead to genomic instability, particularly in telomeres. This has been implicated as a possible cause for cells switching to the alternative lengthening of telomeres (ALT) pathway. *DAXX/ATRX* mutations are common in multiple types of neoplasms, including pancreatic neuroendocrine tumors [35,36]. In pediatric ACC, *ATRX* mutations are the second most common to *TP53* and are found in about one-third of tumors [24]. *MLL4* is also involved in epigenetic regulation. It is a part of an essential family of histone methyltransferases involved in the development of several cancers [37]. The whole-genome analysis performed by Lavoie et al. identified multiple histone methyltransferases and chromatin remodeling genes that harbored mutations, including *ATRX* [21]. Fojo et al. found that alterations in epigenetic regulators were present in over 50% of the tumors in their cohort [20].

Furthermore, DNA methylation status itself has shown prognostic significance in ACC. Barreau et al. found that ACC is hypermethylated compared to adenomas and can be stratified based on the degree of hypermethylation, with hypermethylated tumors associated with decreased overall survival. Within the hypermethylated group, stratification based on CpG Island Methylator Phenotype (CIMP) into CIMP-high and CIMP-low groups could be performed [38]. CIMP status is recognized across multiple cancer types and is associated with silencing tumor-suppressor genes [39,40]. CIMP-high status carried the worst overall survival in the Barreau et al. study, similar to CIMP-high status in other cancer types [38,40]. The ENS@T analysis created four groups of tumors based on methylation status: two corresponded to CIMP-high and CIMP-low patterns and two to non-CIMP patterns [18]. The TCGA study created three groups based on methylation status, which contributed to the development of the Cluster of Cluster (CoC) groups discussed below. Of note, the CoC I, CoC II, and CoC3 groups that resulted from this analysis correlated with CIMP-low, CIMP-intermediate, and CIMP-high patterns, respectively, and similarly were associated with increasingly poor prognosis [19].

Data from the TCGA project, including DNA methylation status, were used by Thorsson et al. to create six different Immune Subtypes (C1-C6) across many types of solid tumors [41]. ACC had the second-lowest median leukocyte fraction of all tumor types in the study and was predominantly the C4 type, which was lymphocyte deplete with an increased macrophage signature. ACC also had a significant proportion of the C3 or “inflammatory” immune subtype. C3 was characterized by high Th1 and Th17 and low proliferative signatures. C3 had a predominance of *TP53* mutations, and C4 carried more *CTNNB* mutations. C4 tumors were associated with a worse prognosis, and as expected, CIMP-intermediate and -high tumors were predominantly the C4 subtype (77% and 89%, respectively). CIMP-low, however, was more mixed, with the most frequent subtype being C3 (50%). C3 tumors were associated with the best prognosis among all the subtypes [41]. This data again underscores the heterogeneity of ACC and the importance of epigenetic changes and alterations in both tumorigenesis and in the characterization of the TME.

Previously, variance in mRNA expression profiles showed value in distinguishing malignant (C1) and benign (C2) tumors, as well as subclassifying C1 tumors into different prognostic groups [42]. The TCGA and ENS@T studies’ analyses supported the correlation between these groups based on transcriptome and outcome [18,19]. mRNA expression data from the TCGA study created four groups based on the expression of steroids and proliferative markers. This resulted in “steroid-phenotype-high” and “steroid-phenotype-low” groups and then each of these showed “plus proliferation” for those tumors with a higher proliferation index. These groups were highly correlated with the previously described C1 subgroups. Specifically, the high steroid phenotype groups in the TCGA study correlated with the C1A subgroup described by de Reynies and were associated with a poor prognosis [19,42]. Additionally, miRNA expression analysis created another means of grouping tumors, with the ENS@T study finding three distinct miRNA-expressing groups. The Mi1 group in this study showed the highest variance from the normal adrenal cortex. The increased expression of the miRNA-506–514 cluster and decreased expression of the *DLK1-MEG3* miRNA cluster were found in this group [18]. The miRNA-506 cluster is best studied in melanoma and is believed to act as a tumor-suppressor [43]. The *DLK1-MEG3* cluster normally is highly expressed in adrenal tissue and includes two additional long noncoding RNAs [18]. Another long noncoding RNA of interest in the ENS@T and TCGA cohorts was *LINC00290.* Recurrent *LINC00290* deletions were found in both of these adult cohorts and, interestingly, have also been identified in a high percentage of pediatric ACC patients from Brazil [18,19,44].

The TCGA study created the aforementioned CoC groups through a comprehensive analysis that combined the four groups created via mRNA expression profiles, DNA methylation, miRNA expression, copy number variation, and protein expression clusters. This stratified tumors into three groups, CoC I, CoC II, and CoC III. This grouping system had significant prognostic value, as patients in the CoC I group had a 7% rate of disease progression and median event-free survival was not reached in the study, whereas in CoC III tumors, disease progression rate was 96% and median event-free survival period was eight months. As this is not practical to perform in a clinical setting, the group created a 68-probe DNA methylation signature from the TCGA analysis that can be extracted from formalin-fixed samples. This signature was able to place tumors into their corresponding CoC grouping with 92% accuracy [19].

Although the pathways mentioned above appear to be involved in ACC, the frequency of individual mutations is highly variable. In addition, despite all we have learned from the comprehensive genomic analyses, there is still a lack of treatment efficacy of identified targets in clinical trials. For example, ACC commonly overexpresses multi-drug resistance 1 (*MDR1*). *MDR1* encodes P-glycoprotein (Pgp), an ABC transporter known to drive cytotoxic chemotherapies out of cells [45]. In vitro studies have demonstrated an improved response of ACC cell lines to cytotoxic therapy with Pgp inhibition, though no in vivo studies have been performed [46]. None of the clinical trials of Pgp inhibitors in combination with cancer chemotherapeutic agents has successfully provided clinically meaningful benefits [47]. Similarly, because most ACCs overexpress *IGF2* and the IGF-I receptor, clinical trials targeting this pathway were performed in patients with advanced ACC. However, the efficacy was poor, as discussed below.

## 3. Current Treatment Approaches

### 3.1. Surgery

From the first descriptions of ACC it has been clear that surgery offers the only chance of survival for these patients. Open adrenalectomy (OA) remains the gold standard, and the National Comprehensive Cancer Network recommends OA for all stages of the disease [48]. However, there are no level I data available comparing OA to laparoscopic adrenalectomy (LA). The retrospective data available have had mixed results, and no strong conclusion can be drawn [49]. None of these are high-quality data, and it is unlikely that a prospective study will be performed due to the rarity of the disease. Therefore, deciding to perform either approach will rely on the individual surgeon and patient factors.

The role of lymph node dissection (LND) in ACC is another topic of debate. It is a difficult topic to study, because on top of the rarity of ACC in general, the rate of LND during adrenalectomy is only between 8–30% in most studies [50,51,52,53]. Current guidelines in both the United States and Europe do recommend performing an LND in all cases of confirmed or suspected ACC [6,48,54]. What constitutes an “adequate” LND is less clear. The most detailed recommendations are from the European guidelines, which recommend removal of all abnormal, periadrenal, and renal hilar nodes at a minimum; and to consider dissection of the superior mesenteric artery, celiac, and periaortic/caval stations. These are weak recommendations based on low quality evidence [54]. American guidelines do not provide any specifics about the extent of LND [48].

There have been mixed results in the retrospective studies evaluating whether patients benefit from LND. A study by Reibetanz et al. using the German ACC Registry in 2012 found that there was a reduced risk of recurrence (HR 0.65, *p* = 0.042) and disease-related death (HR 0.54, *p* = 0.049) in patients who had undergone LND compared to those who did not, while a similar study by Nilubol et al. using the National Cancer Institute’s Surveillance, Epidemiology, and End Results (SEER) database did not find a significant association between LND and disease-specific survival. These studies only included patients who had more than four or five lymph nodes harvested [51,52]. In a 2017 study using the National Cancer Database (NCDB), Panjwani et al. that overall survival was 79% patients who underwent LND with 4 or more nodes removed, while OS was 53% in those with three or fewer removed and that lymph node positivity was an independent predictor of poor overall survival (HR 3.02, *p* = 0.001). As expected, the removal of four or more nodes significantly increased the detection rate of a positive lymph node (30% vs. 16%, *p* = 0.03) [55]. A more recent study using the NCDB found that while the median number of lymph nodes examined after an LND was only two, lymph node positivity was again an independent predictor of worse overall survival (14 vs. 62 months, *p* = 0.01), patients who underwent LND had slightly improved survival (HR 0.82, 95% CI 0.67–0.99), and that 17.5% of clinically node negative disease were upstaged after LND [53]. Altogether, these data show that LND is underperformed in ACC, it provides valuable prognostic information and has the potential to alter the course of treatment, for example in the significant percentage of patients who are upstaged.

Other surgical approaches to ACC include hyperthermic intraperitoneal chemotherapy (HIPEC) and metastectomy. HIPEC involves directly applying heated chemotherapy to the abdomen to destroy any microscopic cancer deposits that have spread outside the original tumor. A recent retrospective study evaluated HIPEC in the prophylactic and therapeutic setting. When HIPEC was used prophylactically at the initial operation for the primary tumor, there was peritoneal recurrence in 1 of 13 patients at a median follow-up of 25 months. Peritoneal recurrence was 68% for recurrent disease at a median follow-up of 30 months [56]. This is similar to the National Institutes of Health (NIH) experience treating recurrent ACC with HIPEC, where recurrence was 70% at a median follow-up of 23 months and 2/9 patients died from disease progression. As with HIPEC in general, there was high morbidity in both studies, ranging from 40–70% [56,57]. More data are needed to make firm conclusions about the use of HIPEC in the prophylactic setting, but the limited data available have shown it is ineffective for recurrent disease. Metastasectomy is considered case-by-case and has small studies in patients with ACC and lung or liver metastases. While there was no apparent increase in survival for these patients, due to the limited availability of effective systemic treatments, metastectomy may be beneficial for select patients [58,59].

### 3.2. Chemotherapy

Traditional chemotherapeutic agents have shown little promise in improving outcomes in ACC. Mitotane, as discussed above, is FDA-approved for patients with locally advanced and metastatic disease. The prophylactic use of adjuvant mitotane in early-stage disease to prevent recurrence is a topic of debate. The ADIUVO trial prospectively studied this question. It randomized 91 patients with early-stage disease between mitotane treatment and observation. There was no significant difference in the primary outcome of recurrence (HR 1.3, *p* = 0.54) or death (HR 2.17, *p* = 0.29). As such, adjuvant mitotane is not recommended for this group of patients [60].

The benefit that patients with advanced stage do obtain from mitotane is also generally marginal, as the most extensive study to date on mitotane monotherapy demonstrated a median progression-free survival of 4 months and overall survival of 18 months [5]. A positive prognostic factor was a mitotane blood level > 14 mg/L, which is considered the therapeutic concentration. This concentration is difficult to achieve and requires frequent blood draws, with some studies showing that only about 60% of patients reach this target concentration [4,5]. In addition, the significant side-effects cause discontinuation rates as high as 30% [4].

One area where there have been promising results with the use of mitotane is in combination with local treatments. Preclinical evidence over the past decade has shown that mitotane sensitizes ACC cells to radiation, likely through inducing cell cycle arrest and inhibiting DNA repair pathways [61,62]. In a recent clinical study, patients with low volume metastatic disease (1 or 2 metastatic lesions) had a progression free survival of 6.8 months when treated with both mitotane and a local therapy for their metastatic lesion(s) compared to a PFS of 3.3 months with mitotane treatment alone (HR 0.39, 95% CI 0.22–0.68). Local therapies used were surgery, interventional radiology techniques, and radiotherapy. There were complete responses in 10 (13%) of the patients treated with mitotane and a local therapy, most of which were surgery or an interventional radiology technique. Because mitotane is the standard of care and was given to all patients in the study, it is difficult to determine the effect of the drug versus the effect of the local treatments. The patients in this trial were similarly plagued by the side-effects of mitotane, with 21% of patients permanently discontinuing treatment and another 23% having an interruption in treatment. In addition, only 60% of patients reaching a blood concentration of >14 mg/L [63]. Though promising, there is still more research needed to define the benefit of mitotane in these patients.

Various other chemotherapeutics have been used in the treatment of ACC. The FIRM-ACT study evaluated etoposide, doxorubicin, and cisplatin plus mitotane (EDP-M) vs. streptozocin plus mitotane and is the only randomized, controlled trial of combination chemotherapy. Though treatment with EDP-M led to a statistically increased overall response of 23% (vs. 9% with streptozocin and mitotane) and prolonged progression-free survival by three months (for a total PFS of 5.6 months), the clinical significance is marginal. There was no effect on overall survival in this study [64]. With no alternatives that significantly impact survival, EDP-M has become the standard of care for unresectable and metastatic disease [6,48]. Second-line chemotherapy in patients with advanced ACC includes gemcitabine-alone or in combination with another agent, such as 5-fluorouracil, capecitabine, vinorelbine, carmustine, or erlotinib. Partial or complete response remains uncommon (less than 10%) [65,66].

### 3.3. Radiation

Radiation therapy (RT) as an alternative method for local control has been investigated in ACC. The American guidelines suggest considering postoperative RT for those at high risk of recurrence after primary surgery (positive margin; Ki-67 >10%; rupture of capsule, large size, or high grade), while European guidelines recommend its use for those with positive margins or unresectable disease [6,48]. In practice, only about 10% of patients in the U.S. receive radiation as part of their treatment [7]. The efficacy of RT as a postoperative adjunct varies widely among studies, and all are retrospective, though most suggest at least some benefit. [67,68,69,70,71,72]. One of the initial studies on the topic from Fassnacht et al. in 2006 using the German ACC Registry showed a decreased local recurrence risk in those who received postoperative RT (12%) compared to those who did not (79%, *p* = 0.01), but there was no difference in overall survival [72]. Similarly, Sabolch et al. found decreased local recurrence risk (5% vs. 60%, *p* = 0.0005) in a U.S. population [70]. Neither study had more than 20 patients in the treatment group [70,72]. More recently, Ginsberg et al. used the NCDB to review 1433 ACC patients at a high risk of postoperative recurrence per NCCN guidelines, of whom 259 underwent adjuvant RT. When the data were adjusted for prognostic variables, they found a significant improvement in OS in those who received postoperative RT (HR 0.68, *p* = 0.001) [71]. Together these studies supports the guideline recommendations and further underscore the small percentage of patients who undergo RT despite these recommendations.

### 3.4. Thermal Ablation

Image-guided thermal ablation using microwave or radiofrequency ablation (RFA) techniques is another local control modality used in multiple cancer types. In ACC, it is best studied in the management of metastasis. In the most substantial study of ablation in ACC to date, thermal ablation initially cleared lung and liver metastases in over 90% of cases. Unfortunately, as is expected with metastatic ACC, disease recurrence was inevitably high. A total of 66% of patients had disease progression and 50% died from cancer-related causes. Patients selected to receive tumor ablation had a five-year survival of 44%, which is higher than the 10–15% five-year survival typically quoted for stage IV disease [4,6,73]. This is likely due, at least in part, to selection bias. This procedure was safe, with only one serious complication and no mortalities [73]. Ablation has not been independently studied for primary ACC, though it has been used for other lesions of the adrenal glands with some success. The technical difficulty of accessing the adrenals poses a challenge for more widespread use. As ablation technology improves and adrenal-specific equipment becomes available, we may see an increase in the use of thermal ablation for all types of adrenal masses. More high-quality data will be needed to determine in which patients it will provide the most benefit [74].

### 3.5. Targeted Therapy

Given our improved understanding of the molecular alterations present in ACC, several small molecule inhibitors targeting these dysregulated molecular pathways in ACC have been tested in preclinical settings and clinical trials. Most of the attempts thus far have been focused on the IGF/mTOR pathway, including a randomized controlled trial with linsitinib (IGF1R inhibitor) monotherapy versus placebo. This study found disease stabilization in only 3 of 90 patients, and there was no improvement in overall or progression-free survival versus placebo [28]. Another trial of 20 patients who received cixutumumab, an anti-IGF1R monoclonal antibody, and mitotane was stopped prior to randomization as most patients were progressing on treatment, with the median PFS being six weeks. A minority of patients did have stabilization of disease on this regimen and there was a single case of a partial response [75]. Doubly targeting this pathway with cixutumumab and sirolimus (mTOR inhibitor) showed disease stabilization in 40% of patients in a phase 1 trial [76]. Targeting the VEGF and EGFR pathways has also been studied in small series, but these studies’ results have been disappointing. None have been able to improve survival outcomes [77,78,79].

### 3.6. Immunotherapy

The efficacy of immunotherapy in patients with advanced ACC remains low. ACC has low immunologic activity, a low tumor mutational burden (TMB), low rates of microsatellite instability, and variable expression of programmed death 1 (PD-1) and its ligand (PD-L1) [80]. The reasons for this are under active investigation but steroid overproduction leading to decreased T-cell activity in the tumor microenvironment (TME) is at least partially responsible [81]. There are small series using PD-1/PD-L1 inhibitors and the cytotoxic T-lymphocyte associated antigen 4 (CTLA-4) for ACC. Across all of these studies, overall response rates are low and none have shown a significant effect on survival outcomes [80]. However, there has been some clinical benefit; about 50% of patients will have a partial response or disease stabilization for some time [17,80,82,83,84]. Clearly, more work needs to be carried out in this area but results thus far have not shown immunotherapy alone to be nearly as effective as it has been in other types of cancer.

### 3.7. Pediatrics

The above discussion focuses on adult ACC, though this is the basis for the treatment approaches in pediatric ACC [13]. Surgery and mitotane remain the cornerstones of treatment. Five-year overall survival after complete surgical resection in stage I disease is over 90% and over 70% in stage II [12,13,85]. Outcomes in stage III disease vary between studies, many with small numbers of patients, but in one study a combination of chemotherapy and surgery in 24 stage III patients led to a five-year OS of 94.7% [9,12,14,85]. For advanced disease, there are limited data on the use of adjuvant mitotane in children, but there is some evidence that it can improve OS if used for over 6 months and a level of >14 mg/L is achieved [85]. Combinations of etoposide, cisplatin, and doxorubicin with mitotane are typically used as (salvage) systemic therapy [12,13]. Radiation is typically not used in pediatric ACC due to the high prevalence of germline *TP53* mutations and the subsequent high risk of secondary cancer development [13]. There are no studies available on the use of novel therapeutics for ACC in pediatrics, though a clinical trial using chimeric antigen receptor autologous T cells (CAR-T) for pediatric patients with solid tumors, including ACC, is underway (Table 1).

## 4. Barriers to Further Progress

The obvious major obstacle in ACC research is the rarity of the disease. With one case/million/year, enrolling patients in studies is slow, and enrollment periods are long despite multi-site enrollment. This can mean that practice patterns have changed or new technologies and medications have emerged between the beginning and end of a study. These are some of the reasons that prospective randomized studies are few and far between for ACC. In addition, for ACC in particular, patients have usually undergone prior treatment with mitotane and other chemotherapeutics before referral to a trial [15,75]. Mitotane pretreatment complicates the picture as the serum concentration is so important for its effectiveness. If these data are available, they likely will show a broad range of concentrations between patients. Mitotane also disrupts the metabolism of many other chemotherapeutic agents via the potent induction of CYP3A4, thereby adding another factor to further muddy the water [79,86].

With all of these difficulties and the long time needed for clinical studies of ACC, there is added importance to having high-quality in vitro models. As discussed, ACC’s molecular profile, steroid expression, and response to treatment are all very heterogenous, meaning that a single cell line cannot possibly represent the spectrum of disease. Unfortunately, until recently, there was only one cell line available, NCI-H295, which was developed in the 1980s [87,88]. Another line is SW-13, though this line does not produce steroids, and it is unknown whether the source of the adrenal tumor was a primary adrenal malignancy or metastasis from another site [89,90].

The recent development of five new ACC cell lines: CU-ACC1, CU-ACC2, MUC-1, JIL-2266, and TVBF-7, marks substantial progress in this field. Table 2 lists the steroid expression and mutation profiles for commonly implicated genes for these cell lines. Importantly, there is considerable variability between these cell lines regarding steroid expression and the mutations each harbor. Additionally, CU-ACC1, CU-ACC2, MUC-1, and TVBF-7 were transferred from metastatic lesions, and all of the new lines except CU-ACC1 were exposed to systemic therapy before harvesting [91,92,93,94,95]. Pre-treated and metastatic lines are a higher fidelity in vitro model for advanced disease than was previously available. It is important to note that CU-ACC2 and JIL-2266 have mutations in mismatch repair genes, improving our ability to study this subset of patients [93]. However, only NCI-H295R and SW-13 are commercially available.

Patient-derived organoids (PDOs) provide another means for in vitro study of disease in a more personalized fashion. Fresh patient tissue samples from biopsy or resection are taken to the laboratory, where a three-dimensional culture of organoids containing multiple cell types can be created [96]. PDOs maintain many of the original tumor’s genomic and transcriptomic elements in culture, and their treatment response in vitro has been shown to correlate with the patient’s in vivo response to the same treatment [97,98].

While PDOs hold much promise for the individualization of cancer treatment, significant limitations still need to be overcome. While PDOs are multicellular and represent some of the structural elements of the primary tumor, the conventional methods of creating PDOs do not grow blood vessels or immune cells and therefore do not truly mirror the in vivo TME [99]. Establishing a PDO from an individual patient also presents barriers. Fresh tissue samples are more difficult to obtain in the clinical setting, and for metastatic patients, biopsy specimens are typically the only available tissue [96]. In a prospective trial of creating PDOs for personalized treatment of colorectal cancer, a period of 10 weeks was needed, on average, from obtaining the specimen to returning drug screening results. In addition, the success rate of establishing a PDO was only 58% [100]. Currently, there are no established PDO response criteria defining “sensitive” when performing drug screens. This will need to be determined for individual tumor types and may differ between individual patients or samples taken from the same patient at different times or locations [98]. In the context of ACC, there has only recently been one report of the successful creation of a PDO from an adrenocortical carcinoma [101].

Using xenografts is one way to overcome many inherent problems with in vitro models [102]. Xenografts using established cell lines shed light on how these cells grow in an environment more similar to a natural tumor, but the inability to represent the full spectrum of disease limits their potential [103]. Patient-derived xenografts (PDXs) are a way to improve upon the lack of heterogeneity in cell lines and, similarly to PDOs, represent a promising approach to personalized medicine [104]. Creating a PDX involves grafting fresh tumor samples into an immunodeficient mouse, either subcutaneously or orthotopically, at the corresponding site of the original tumor. Three adult and one pediatric PDX models for ACC have been developed. The pediatric PDX, SJ-ACC3, was created from the tumor of an 11-year-old male with a germline *TP53* mutation and is the only available preclinical model for pediatric ACC. There was no associated cell line derived from this xenograft [105]. The adult PDX models, MUC-1, CU-ACC1, and CU-ACC2 all have associated cell lines for in vitro study [91,92].

These models are an improvement as they allow for the study of the tumors and their interactions with surrounding vasculature and connective tissue, but they are very time-consuming and labor-intensive to create. Successful engraftment ranges from 15–85%, depending on the tumor type [104]. In addition, grafted tumors have TME interactions, but this is not a perfect re-creation of the native TME, especially for PDXs grafted subcutaneously. Orthotopic transplantation better mirrors the native TME; however, it is an even more labor-intensive process, and tracking tumor growth is challenging [102].

Transgenic mouse models have also been developed for the study of ACC. As the IGF2, Wnt-signaling, and p53/Rb pathways are frequently over-expressed in ACC, transgenic mice for ACC have focused on these pathways. Mice over-expressing *IGF2* and *CTNNB1*, alone and in combination, did not reliably develop ACC [103]. Inhibiting the p53/Rb pathway with the Simian Virus 40 (SV40) oncogene under the control of the adrenal cortex-specific promoter, however, was successful in causing the development of metastatic ACC [106]. Another transgenic mouse with *p53* loss of function and *CTNNB1* gain of function led to ACC development in over 80% of mice at 12 months [107]. While there are differences in human and mouse biology, the ability to reliably have ACC develop without needing grafting provides a much better representation of TME interactions in vivo.

Though there is much potential in these models, there is still a lack of diversity among them. The three PDXs are the same as three of the cell lines, and there are only two transgenic mouse lines that reliably develop ACC. This is even more of a problem for pediatric ACC, as there is only one PDX and no cell lines available for in vitro study. More diversity is needed for the adequate representation of both adult and pediatric ACC in the preclinical setting.

## 5. Knowledge Gaps

### 5.1. Manipulation of Which Molecular Signature Will Lead to a Breakthrough?

Over the past few decades, there have been great successes in treating other types of cancer by identifying and effectively targeting a “hallmark mutation.” As discussed, the recurrent mutations in ACC, such as *CTNNB1*, *p53*, *Rb*, *ATRX/DAXX*, and *TERT*, provide some insight into the disease development but few are targetable. Further, the frequency of an alteration in any single gene is rarely greater than 10–20%. Even considering all the genes in a pathway, no pathway is altered in >50% of patients [18,19].

These findings support the idea that mutation analysis alone is not likely to fully explain this complex disease. The frequency with which epigenetic and noncoding RNA alterations have been identified points to these areas as promising targets for future research. The finding of a recurrent deletion in the long-noncoding RNA *LINC000290* in both the TCGA and ENS@T cohorts is one example of a potential target that has not yet undergone further study [18,19]. The importance of long noncoding RNA in cancer development is increasingly being recognized as targets for therapy, but there is still a paucity of research available for their practical use [108].

Identifying miRNA expression patterns involved in pathogenesis has led to further insight across tumor types. To date, these patterns have been most helpful in determining prognosis in ACC. The most notable example comes from a retrospective study where miR-483-5p, an miRNA found within an intron of the *IGF2* gene was measured in the serum of patients with ACC and found to have 100% specificity for predicting postoperative recurrence within three years [109]. miRNA clusters were also valuable in the TCGA analysis in creating their Cluster of Clusters, which were strongly associated with disease progression rates [19]. Various other miRNAs have been implicated in ACC whose known roles span numerous pathways, including cell cycle regulation, mTOR, and Akt, among others. Interestingly, Wnt-signaling and p53-Rb have not been tied to miRNA expression profiles thus far [108]. Much research is still needed to further delineate miRNA’s roles in ACC and whether they represent a useful therapeutic target.

Epigenetic alterations are another recurring theme in these studies. DNA methylation profiles have demonstrated significant value as prognostic markers, and multiple genomic studies implicate genes involved in epigenome maintenance [18,19,20]. Studies on methylation profiles have uncovered thousands of genes affected by hypermethylation in ACC. For example, in one study, 3325 genes were identified as hypermethylated in CIMP-high tumors compared to non-CIMP tumors. Genes with various functions, including some tumor suppressors (G0S2, PLAGL1, and NDRG2), were among those affected [38]. While this has expanded our knowledgebase for ACC, the specific role of these genes has yet to be elucidated and it has underscored the diversity of molecular profiles in ACC. The specific role of these epigenetic changes in the development and spread of ACC is another area requiring much more research.

The preceding data all come together to emphasize that it is unlikely that a one-treatment-for-all approach will succeed in ACC. These diverse molecular alterations in ACC highlight the importance of personalized treatments based on the tumor’s molecular profile. Further work is needed to determine the pathogenic molecular mechanisms in ACC initiation and progress to identify novel therapeutics.

### 5.2. How Can the Microenvironment Be Manipulated to Stop Tumor Growth?

Another significant gap in the current knowledge is the interactions in the TME that allow malignant cells to escape the immune system. The importance of PD-1/PD-L1-mediated T-cell deactivation in other cancers and the success of PD-1 inhibition in their treatment has not played out in ACC. In other areas where immunotherapy has been successful, such as melanoma, PD-1 expression and response typically correlate [110,111]. Interestingly, an association has not been found between the level of PD-1 expression and those who achieve stabilization or a partial response in ACC [82,83,84]. A better understanding of why this is not the case for ACC will be essential for successfully targeting the PD-1/PD-L1 axis in ACC. It will primarily help determine which other pathways can be targeted in conjunction with PD-1 to obtain a more robust response. The only data available are from a Phase II trial evaluating a combination of CTLA-4 inhibition (ipilimumab) with PD-1 inhibition (nivolumab) in patients with rare urogenital tumors. The primary endpoint was the objective response rate determined by RECIST criteria [112]. For ACC patients treated in this study, eight of sixteen progressed, seven had a stabilization of disease, and only one patient showed any objective response [113]. The available data do not provide much optimism for the effectiveness of these drugs; however, there will be more information on this in the future, as multiple trials are underway with PD-1 inhibitors (Table 1).

CD276 (B7-H3) is another molecule over-expressed in multiple cancer types and is implicated in evading immune surveillance. It directly inhibits natural killer (NK) cell function, and the inhibition of CD276 increases T-cell activity in the TME [114]. Over 90% of ACC expressed CD276 on the cell membrane or in the cytoplasm of ACC cells or tumor-associated vascular cells. Furthermore, unlike PD-1, increased CD276 expression correlates with higher T stage and advanced ENS@T stage and is independently associated with shorter overall (HR 2.8) and recurrent-free survival (HR =7.52) [115]. A phase 1 study is ongoing using CD276 chimeric antigen receptor autologous T-cell therapy for pediatric patients with solid tumors (including ACC), but otherwise, there is no clinical data on CD276 inhibition in ACC. Further studies are needed to elucidate the role of CD276 in ACC progression and the efficacy of anti-CD276 in patients with advanced ACC.

Chemokine signaling is also an important part of the tumor microenvironment. The chemokine CXCL12 and its receptors CXCR4 and CXCR7 are involved in how tumor cells are released from the primary tumor and find their metastatic sites. This likely occurs via the lower expression of CXCL12 in the primary tumor and high expression outside the tumor, creating a gradient that precipitates invasion and spread [116]. Increased intratumoral CXCL12 is also believed to help recruit T-cells to the TME and improve local control of tumor cells. This axis has recently been studied in ACC which showed that high CXCL12 expression correlated with improved DFS (81.9 vs 24.1 months, *p =* 0.01, PFS (81.9 months vs. 24.6 months, *p =* 0.01), and risk of recurrence or metastasis (HR 0.12, *p =* 0.04) [116]. In vitro and preclinical in vivo studies of this axis with rosiglitazone, known to inhibit the downstream signaling of CXCR4, caused the increased expression of CXCL12 in the tumor cells and decreased invasion [117]. This pathway is an important element of the TME that warrants further study and consideration as a target for clinical use.

A unique element of the microenvironment for adrenal tumors, including ACC, is the relative lack of tumor-infiltrating lymphocytes (TILs) in the tumors or surrounding retroperitoneal fat. Hypothesized causes for this include alterations in chemokine signaling and the presence of glucocorticoids in the TME [17]. It is well-known that steroids down-regulate the immune system and, in particular, significantly decrease the activity of the cytotoxic T-cells. Patients with Cushing’s syndrome from ACC have shorter overall survival but intratumoral glucocorticoid excess may be present even without clinically apparent Cushing’s syndrome [81,118]. The dampening of T-cell activity by steroids is likely a significant factor in allowing tumor cells to evade the immune system in ACC [80]. A recent study on the relationship between glucocorticoids and TILs showed that an increased concentration of TILs is an independent predictor of improved ACC prognosis. Further, the presence of excess glucocorticoids significantly worsens prognosis, and the presence of steroids combined with low concentrations of TILs predicts the worst prognosis [81]. The effects of glucocorticoids in the TME and systemic immunosuppression in ACC-related Cushing’s syndrome are likely factors as to why the response to immunotherapy has been disappointing thus far in ACC. Counteracting these effects may improve this response, and there is a clinical trial underway to investigate the effects of a PD-1 inhibitor (pembrolizumab) combined with a glucocorticoid receptor antagonist (relacorilant, Table 1).

## 6. Addressing the Knowledge Gaps

### 6.1. Laboratory Research

Over the past decade, numerous advances in the preclinical models have been made available for the study of ACC. The newly developed cell lines provide a broader range of ACC phenotypes for cell culture experiments, but, as shown in Table 2, there is still limited heterogeneity among these lines when compared to what we know about the diversity of the disease [18,19,91,92,93,94]. PDOs are a new method with immense potential to create a personalized treatment approach. If PDOs in ACC maintain the genetic and epigenetic signatures of the native tumor as in other cancer types, uncovering these changes for an individual may help elucidate a targetable molecular signature [97]. As PDOs can be generated from both primary tumors and metastases, they can help elucidate the tumor biology for advanced disease, which is known to be even more heterogeneous and harbor more mutations [119]. There is much work to be carried out before this becomes a reality. Prerequisites to the clinical utility of PDOs include verifying the PDO’s genetic and epigenetic fidelity to the primary tumor, finding candidate drugs to target distinct molecular signatures, and determining what qualifies as a treatment response in the PDO [99].

Addressing our limited knowledge about the tumor microenvironment will require in vivo models. Although few in vivo models are available, they are the best ways we have to represent the TME preclinically. Transgenic mice that develop ACC in their natural environment allow us to study tumor progression over different stages in tumor development, chemokine signaling interactions, and the immune system’s role in the TME. To improve the study of the immune system interactions in the TME in these models, a chimeric mouse model with a human immune system was created and then grafted with the CU-ACC2 PDX [92]. This enabled the study of immune cells in the TME and the effect of PD-1 inhibition on these cells and the tumor itself. In this model, PD-1 inhibition delayed tumor growth and caused increased TIL presence and activity in the TME. Further, pembrolizumab induced a partial response in the patient from which the CU-ACC2 PDX was derived [120]. Although this is an encouraging first study using a humanized mouse model, the CU-ACC2 tumor does not produce steroids and has mutations in the mismatch repair pathway, which is only the case in a fraction of ACC patients [19,92]. Similar studies using this novel approach in the other available models for further characterization of the TME.

### 6.2. Translational Research

Because there are no effective systemic treatment options in patients with advanced ACC, the focus of translational scientific research in ACC should be on identifying novel and effective treatment strategies tailored to a specific molecular profile of ACC. With improved insight into molecular pathophysiology in ACC and the prognostic value of several biomarkers, treatments that target these dysregulated pathways are being studied preclinically [15]. In addition, treatment resistance in ACC is a common problem. The lack of efficacy of chemotherapy in ACC has been associated with an overexpression of the multi-drug resistance P-glycoprotein [45]. Successful personalized treatment will likely rely on broadly treating multiple pathways with combination therapy. Much of the clinical work currently being carried out on combination therapies is with immunotherapy, specifically PD-1 inhibitors (Table 1). These have yet to show overwhelmingly positive results so far, and, as outlined above, there are many other options we have learned about that may lead to more promising results.

One method for studying these in a cost and time efficient manner is quantitative high throughput drug screening (qHTS). This method screens compounds already in use clinically for other indications for their activity against ACC cell lines. qHTS offers several advantages over traditional approaches. The safety profiles and pharmacodynamics of these drugs are known, allowing for the bypass of the early stages of clinical trials, and trials to test the drug’s efficacy can start from the outset. In addition, qHTS screens numerous compounds that would not be tried otherwise, and their mechanism of action may provide new insight into the development of ACC. This can be particularly useful for finding combinations of drugs that may have a synergistic effect [121].

Our group has put qHTS to use for identifying new treatments for ACC. We screened thousands of compounds for activity against ACC cell lines and found multiple new potentially active treatments. One example of a novel therapeutic option found through qHTS is the combination of the CDK inhibitor flavopiridol and proteasome inhibitor carfilzomib. These had previously shown a synergistic effect in leukemia treatment, and a similar effect was seen in ACC [122]. Because these drugs are already FDA-approved, developing a clinical trial can be expedited using this combination of therapy [15]. Another compound found via qHTS is the anti-helminth agent niclosamide, which was found to be more effective in inhibiting tumor growth in vitro than mitotane, etoposide, cisplatin, or streptozocin. A possible mechanism for this is inhibition of the Wnt-signaling pathway. Niclosamide has been in use clinically for decades and is known to have a low side-effect profile. qHTS has identified many more compounds, and further work is needed to validate them [122,123]. As discussed above, the critical barrier to the preclinical research of novel therapeutics is the lack of diversity in preclinical models.

### 6.3. Clinical Trials

The main challenge in conducting clinical trials in patients with rare cancers, such as ACC, is patient enrollment. The level I data derived from a prospective randomized clinical trial inevitably requires multi-center enrollment. This highlights the importance of collaboration among researchers from multiple institutions with different strengths to achieve a common goal of improving treatment outcomes in patients with ACC. Table 1 shows the ongoing clinical trials and their underlying target pathways. Based on what we know from prior clinical data on PD-1 or VEGF inhibitor monotherapy, these trials are unlikely to yield a significant improvement over the current treatments available for ACC. An interesting ongoing trial is using nivolumab combined with a therapeutic vaccine (EO2401), which has been shown to activate the immune system against cancer antigens in vitro. Recently published preliminary results showed disease stabilization rates of 60%, but objective response rates are still underwhelming at 24% [124]. As discussed throughout this paper, there are many other options based on promising preclinical studies to explore in treating ACC and more clinical trials are needed.

## 7. Conclusions

There has not been a substantial breakthrough in the treatment of ACC over the past 50 years. Despite a vastly improved understanding of the disease over the past few decades, patients with advanced ACC still have a dismal prognosis with current treatment options. The availability of new cell lines and the utilization of innovative approaches, such as qHTS, may hold the key to finding a more effective treatment. Until the time comes when we can sample a patient’s tumor and provide the specific treatment for that individual, combination therapy is likely to be the next step forward in the management of ACC. The tools needed to find a treatment that works are available, and with continued work in the laboratory, translational, and clinical settings, there is much promise for a breakthrough.

## Figures and Tables

**Table 1 cancers-14-05245-t001:** Ongoing clinical trials for the treatment of ACC (from the ClinicalTrials.gov database).

Name	Phase	Drug	Target	Status	Location
Cabazitaxel Activity in Patients With Advanced Adrenocortical-Carcinoma Progressing After Previous Chemotherapy Lines (CabACC)	II	Cabazitaxel	N/A	Unknown	Italy
Cabozantinib in Advanced Adrenocortical Carcinoma (CaboACC)	II	Cabozantinib	VEGF	Active (not recruiting)	England
Adjuvant Chemotherapy vs. Observation/Mitotane After Primary Surgical Resection of Localized Adrenocortical Carcinoma (ACACIA)	III	Mitotane	N/A	Unknown	Italy
A Novel Therapeutic Vaccine (EO2401) in Metastatic Adrenocortical Carcinoma, or Malignant Pheochromocytoma/Paraganglioma (Spencer)	I/II	EO2401 (Vaccine) + nivolumab	IL13Ra2, BIRC5 and FOXM1 (vaccine), PD-1	Recruiting	International
Surgery and Heated Intraperitoneal Chemotherapy for Adrenocortical Carcinoma	II	HIPEC	N/A	Recruiting	U.S.
Cabozantinib (VEGF) in Treating Patients With Locally Advanced or Metastatic Unresectable Adrenocortical Carcinoma	II	Cabozantinib	VEGF	Active, not recruiting	U.S.
Mitotane With or Without Cisplatin and Etoposide After Surgery in Treating Participants With Stage I-III Adrenocortical Cancer With High Risk of Recurrence (ADIUVO-2)	III	Mitotane ± cisplatin + etoposide	N/A	Recruiting	U.S., Poland
Nivolumab Combined With Ipilimumab for Patients With Advanced Rare Genitourinary Tumors	II	Nivolumab + ipilimumab	PD-1, CTLA-4	Recruiting	U.S (Multiple sites)
Cabozantinib-S-Malate in Treating Younger Patients With Recurrent, Refractory, or Newly Diagnosed Sarcomas, Wilms Tumor, or Other Rare Tumors	II	Cabozantinib	VEGF	Active (not recruiting)	U.S. (multiple sites)
Pembrolizumab in Treating Patients With Rare Tumors That Cannot Be Removed By Surgery or Are Metastatic	II	Pembrolizumab	PD-1	Active (not recruiting)	U.S. (MD Anderson)
Nivolumab and Ipilimumab in Treating Patients With Rare Tumors	II	Nivolumab + ipilimumab	PD-1, CTLA-4	Recruiting	U.S. (multiple sites)
A Phase 1/1b First-In-Human, Dose-Escalation Study to Evaluate the Safety, Tolerability, Pharmacokinetics, and Pharmacodynamics of IPI-549 (eganelisib) Monotherapy and in Combination With Nivolumab in Subjects With Advanced Solid Tumors	I	Eganelisib + nivolumab	mTOR/PI3K, PD-1	Active (not recruiting)	U.S. (multiple sites)
B7-H3-Specific Chimeric Antigen Receptor Autologous T-Cell Therapy for Pediatric Patients With Solid Tumors (3CAR)	I	Autologous T cells	B7-H3 (CD-276)	Recruiting	U.S.
Cisplatin-Based Chemotherapy and/or Surgery in Treating Young Patients With Adrenocortical Tumor	III	Cisplatin	N/A	Active (not recruiting)	U.S (multiple sites)
Phase II Trial of Pembrolizumab Plus Lenvatinib in Advanced Adrenal Cortical Carcinoma (ACCOMPLISH)	II	Pembrolizumab + Lenvatinib	PD-1, VEGF	Not yet recruiting	
A Phase II Clinical Trial of Single Agent Pembrolizumab in Subjects With Advanced Adrenocortical Carcinoma	II	Pembrolizumab	PD-1	Active (not recruiting)	U.S.
Phase II Study for Combination of Camrelizumab and Apatinib (VEGF) in the Second-line Treatment of Recurrent or Metastatic Adrenocortical Carcinoma	II	Camrelizumab + apatinib	PD-1, VEGF	Not yet recruiting	
Study of Relacorilant in Combination With Pembrolizumab for Patients With Adrenocortical Carcinoma Which Produces Too Much Stress Hormone	I	Pembrolizumab, relacorilant	PD-1, glucocorticoid receptor	Recruiting	U.S.

**Table 2 cancers-14-05245-t002:** Currently available cell lines for the study of ACC. GOF = gain of function. A (−) means there is no cortisol production. A (+) indicates cortisol production, and multiple (+) signs approximate increased cortisol production compared to the NCI-H295 cell line.

Cell Line	Year Published	Origin	Treatment Prior to Culture	Cortisol Production	Harbored Mutations	Doubling Time
SW-13	1973 [90]	Small cell tumor, origin debated	Not recorded	−	TP53	24 h
NCI-H295R	1990 [87]	Primary	none	+	TP53, CTNNB1 (GOF)	24–36 h
MUC-1	2016 [91]	Metastasis (neck, subcutaneous)	EDP-M	++	TP53	Not published
CU-ACC1	2018 [92]	Metastasis (renal)	none	++++	CTNNB1 (GOF)	35 h
CU-ACC2	2018 [92]	Metastasis (liver)	Radiation, mitotane	−	TP53, MSH2	29 h
JIL-2266	2021 [93]	Primary	Mitotane, metyrapone	+/− (dependent on media)	TP53, MUTYH	41 h
TVBF-7	2022 [95]	Metastasis (peri-renal lymph node)	EDP-M	+++	APC	Not published

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
