# Peer review of "Bridging the Scientific Gaps to Identify Effective Treatments in Adrenocortical Cancer"

_cancers, 2022, doi:10.3390/cancers14215245_

Round 1

Reviewer 1 Report

# General comments

Michael & Nilubol present a comprehensive review on current treatment options and molecular understanding of ACC, suggesting knowledge gaps to overcome to treatment. The manuscript is well written and interesting to read.

# Comments for revision:

The Authors dedicate a large space of the manuscript to discussing the molecular landscape of ACC (at Section 2), summarizing important findings from the TCGA-ACC and ENSAT studies. This section seems particularly important to provide the reader with background information for Section 3 (Current Treatment Approaches), specially about the immunotherapy discussion. In order to improve this background information, I suggest also discussing the ACC immune subtypes defined by Thorsson et al. (Immunity, 2018; doi: 10.1016/j.immuni.2018.03.023). This landmark study provided new insights into the immune landscape of cancers using TCGA data, including ACC. I also suggest further discussion on the ACC's mRNA classification defined by Zheng et al. (Cancer Cell, 2016; doi:10.1016/j.ccell.2016.04.002); for example, although these authors have introduced the TCGA-ACC cohort by highlighting the integrative molecular subtypes, they also present molecular subtypes for each data type, including classification of steroid phenotypes.

Author Response

The Authors dedicate a large space of the manuscript to discussing the molecular landscape of ACC (at Section 2), summarizing important findings from the TCGA-ACC and ENSAT studies. This section seems particularly important to provide the reader with background information for Section 3 (Current Treatment Approaches), specially about the immunotherapy discussion.

Point 1: In order to improve this background information, I suggest also discussing the ACC immune subtypes defined by Thorsson et al. (Immunity, 2018; doi: 10.1016/j.immuni.2018.03.023). This landmark study provided new insights into the immune landscape of cancers using TCGA data, including ACC.

Response 1: Thank you for your valuable suggestions. We have included this paper and its relevance to ACC in the background section. Please see lines 181-194.

Point 2: I also suggest further discussion on the ACC's mRNA classification defined by Zheng et al. (Cancer Cell, 2016; doi:10.1016/j.ccell.2016.04.002); for example, although these authors have introduced the TCGA-ACC cohort by highlighting the integrative molecular subtypes, they also present molecular subtypes for each data type, including classification of steroid phenotypes.

Response 2: Thank you for this suggestion. We have updated our discussion of the specific mRNA classifications and steroid phenotypes in ACC from the paper by Zheng, et al. Please see lines 198-205.

Reviewer 2 Report

Thank you for the nice review regarding such an important issue in the context of ACC. This review is nice written, however some topics should e adressed more in detail:

1.) radiotherapy: up to my knowledge over the last years there is a open discussion about a benefit of radiotherapy and there are different studies /reviews, which could show a better OS. I Think  this section should be discussed more in detail and more differenciated

2.) surgery: the potential role of lymphadenectomy should be discussed in addition

3.) the entity in pediatric patients has not discussed at all and should be included

Author Response

Thank you for the nice review regarding such an important issue in the context of ACC. This review is nice written, however some topics should e adressed more in detail:

Point 1: radiotherapy: up to my knowledge over the last years there is a open discussion about a benefit of radiotherapy and there are different studies /reviews, which could show a better OS. I Think  this section should be discussed more in detail and more differentiated

Response 1: Thank you for your comments and suggestion. We have revised the radiotherapy accordingly. Please see lines 352-371.

2.) surgery: the potential role of lymphadenectomy should be discussed in addition

Response 2: Thank you for your suggestion. A discussion of the role of lymphadenectomy has been added. Please see lines 251-283.

3.) the entity in pediatric patients has not discussed at all and should be included

Response 3: Thank you for your suggestion. Notable areas where pediatric ACC differs or is similar to adult have been added throughout the paper, as well as a discussion of the treatment of ACC in pediatrics. Please see lines 102-111 and 424-438 for the most significant changes.